# Consumer Awareness and Acceptance of Biotechnological Solutions for Gluten-Free Products

**DOI:** 10.3390/foods12091808

**Published:** 2023-04-26

**Authors:** Paola Sangiorgio, Simona Errico, Alessandra Verardi, Silvia Massa, Riccardo Pagliarello, Carla Marusic, Chiara Lico, Ombretta Presenti, Marcello Donini, Selene Baschieri

**Affiliations:** 1Laboratory Bioproducts and Bioprocesses, ENEA, Italian National Agency for New Technologies, Energy and Sustainable Economic Development, Trisaia Research Centre, 75026 Rotondella, Italy; 2Laboratory Biotechnologies, ENEA, Italian National Agency for New Technologies, Energy and Sustainable Economic Development, Casaccia Research Centre, Santa Maria di Galeria, 00123 Rome, Italy

**Keywords:** gluten-free products, public opinion, consumer behavior, consumer acceptance, food biotechnology

## Abstract

Celiac disease is an immune-mediated disorder caused by the ingestion of gluten proteins. The gluten-free diet is currently the only therapy to achieve the symptoms’ remission. Biotechnological approaches are currently being explored to obtain safer and healthier food for celiacs. This article analyzes consumer awareness and acceptance of advanced biotechnologies to develop gluten-free products. An online snowball sampling questionnaire was proposed to 511 Italian participants, selected among celiac and non-celiac people, from December 2020 to January 2021, during the second wave of the COVID-19 pandemic. Overall, 64% of respondents favor food biotechnology, as long as it has benefits for health or the environment. Moreover, biotechnology perception differs according to education level and type. A total of 65% of the survey participants would taste gluten-free products obtained through a biotechnological approach, and 57% would buy them at a higher price than the current market price. Our results show a change in public opinion about the usefulness of food biotechnology and its moral acceptability compared to 20 years ago. However, the study of public opinion is very complex, dealing with individuals with social, economic, and cultural differences. Undoubtedly, the scientific dissemination of genetic biotechnologies must be more effective and usable to increase the level of citizens’ awareness.

## 1. Introduction

Celiac disease is an immune-mediated pathology that affects about 1% of the population in Europe. It is characterized by a state of chronic inflammation of the small intestine triggered, in genetically predisposed subjects, by ingesting gluten proteins contained in the grains (seeds) of certain cereals, including wheat [1]. Complete and permanent elimination of gluten from the diet is the only treatment currently available for the remission of symptoms and to prevent complications. Although considerable progress has been made to improve the palatability of gluten-free foods, commercial products are usually high in calories and expensive. However, gluten-free diets have become increasingly popular in recent years among the general population, athletes, and patients with clinical conditions other than celiac disease, including non-celiac gluten sensitivity, irritable bowel syndrome, neurological diseases, and autism [2].

The replacement of the gluten mesh is currently one of the biggest challenges in food technology [3]. In the grain, gluten proteins serve as an energy reserve and support germination during the initial stages of plant development. When the grains are milled, and water is added to the flour to produce dough, the matrix formed by gluten proteins around starch granules turns into an elastic, viscous network. In the dough, during baking, the gluten network slows down the absorption of water by the starch, giving the product tenacity and elasticity, while in bread, it retains the gas bubbles produced by yeast. This property, combined with those of cohesion, homogeneity, visco-elasticity, and tenacity, results in a soft and elastic product that is pleasing to the palate [4]. Gluten proteins, which are numerous and very diverse, are encoded by several genes grouped in loci distributed on different chromosomes and are classified into two groups, gliadins and glutenins [5]. This genetic complexity makes it impossible to generate gluten-free cereals using classical genetic techniques (such as plant breeding) [6].

Patents focused on gluten-free products and ‘celiac disease’ are numerous, and many of them focus on the development of (i) drugs [7]; (ii) chemical-physical or enzymatic treatments for the degradation of gluten in food [8,9]; (iii) non-transgenic low-gluten wheat with CRISPR/Cas9 [10]; and (iv) detoxified gluten proteins through recombinant DNA-based approaches [11].

Biotechnologies, in particular genetic technologies, can help to obtain gluten-free products with characteristics similar to those containing “natural” gluten, as demonstrated by the above studies. However, has public opinion on these technologies improved compared to twenty years ago [12]? Recent studies show that the debate on the use of biotechnology in the agri-food sector is still heated, despite the unanimous chorus of scientists worldwide on their safety and usefulness. However, these studies acknowledge a more favorable public sentiment toward biotech products, especially in relation to genetically modified products [13]. The 2019 European Food Safety Authority survey observed a decrease in the percentage of Europeans choosing GMOs as a food safety concern from 66% in 2010 to just 27% in 2019 [14]. Does this mean that common knowledge has increased?

The purpose of this article is to identify if there has been a positive trend in the perception of biotechnologies and, in particular, of foods produced with biotechnological approaches. Through an online survey of 511 consumers selected by snowball sampling, we studied consumer awareness and acceptance of the use of advanced biotechnologies for gluten-free products. We also evaluated the degree of knowledge of celiac disease on the part of consumers and their propensity to purchase innovative products with detoxified gluten. Finally, we compared our findings with data from similar studies in the literature.

## 2. Materials and Methods

### 2.1. Survey Design

An online snowball sampling questionnaire was used to reach numerous consumers during the second wave of COVID-19 in Italy. During this period, the pandemic crisis has imposed restrictions (stay-at-home policy) to contain the virus’s spread in Italy. In this situation, this non-probability survey sample selection method made it possible to obtain answers quickly from a good number of consumers, guaranteeing the representativeness and generalizability of the sample. The survey was conducted from December 2020 to January 2021 using an ad hoc questionnaire linked to the Google Forms platform. Selected participants were asked to share the invitation with colleagues/friends/family who might be suitable for this study. The target of consumers believed to have the characteristics of interest was reached through the numerous referrals of the initially sampled participants to other potential subjects [15]. A total of 511 Italian participants answered the questionnaire disseminated using different media channels, including personal social networks (e.g., Facebook, WhatsApp, etc.), institutional mailing lists, and specifically dedicated pages.

In this way, we were also able to reach a large proportion of consumers directly or closely affected by the coeliac condition. This is confirmed by the percentage of participants with coeliac disease or family members with coeliac disease: 19% of consumers surveyed compared to 1% of coeliacs on average in Italy and Europe [1].

### 2.2. Questionnaire Design

Particular attention was paid to formulating simple, clear, and concise questions and ordering their succession to lead the consumer by the hand to the heart of the survey. The questionnaire includes various sections:Presentation of the questionnaire and informed consent;Consumer details: province of residence, gender, age (by age group), description of the family unit, educational qualification, level and type of culture, and profession;Purchasing habits: factors that influence their choices;Propensity for novelties in the food field;Degree of knowledge of biotechnologies and their perception;Celiac disease: direct or indirect knowledge of people with celiac disease, degree of knowledge of the celiac disease;Gluten-free products: characteristics and satisfaction;Propensity to consume and purchase products containing detoxified gluten.

The questionnaire was strictly anonymous. The answers were anonymous and confidential: the results were reported aggregately and archived securely so as not to disclose information about individual consumers. The data access was limited to a pool of authorized and identified persons of the Italian National Agency for New Technologies, Energy and Sustainable Economic Development (ENEA). Participation in the survey was voluntary. It was possible to withdraw at any time by closing the browser and not submitting the form. Finally, it was requested to give electronic informed consent to proceed with the answers to the various questions.

### 2.3. Statistical Processing of Data

The survey results were statistically processed using GraphPad Prism version 8 (GraphPad Software, San Diego, CA, USA). Data are shown as means ± standard deviation (SD). In the case of a normal distribution of data of groups, we conducted a one-way ANOVA analysis of variance with multiple comparisons to evidence differences between groups. Tukey’s post hoc test was used as a statistical hypothesis testing, applying a 95% confidence level (*p* < 0,05). In the case of non-normal data, we performed the Kruskal–Wallis test (*p* < 0.05), followed by the Dunn test for multiple comparisons of groups. Significant differences were shown as asterisks on respective graphs, where * *p*-value ≤ 0.033, ** *p*-value ≤ 0.002, and *** *p*-value ≤ 0.001. We used the Chi-square test of independence (confidence interval = 95%) on some data to determine whether categorical or nominal variables are likely to be related.

## 3. Results

### 3.1. Consumer Characteristics

Figure 1 shows the results of the survey relating to consumer characteristics, such as region (or province) of origin, gender, age group, number of family members and minor children, education, and employment.

Survey participants are mainly aged between 18 and 24 (24%), 45 and 54 (23%), and 25 and 34 (19%), of which 65% are female.

Regarding the place of origin, the most represented region is Lazio (206 respondents from Lazio, of which 193 were from Rome), followed by Sicily (56), Lombardy (51), Calabria (42), and Basilicata (34). It is interesting to note that the surveyed sample is balanced between large and small cities.

Regarding the level and type of education, 66% of the participants have a tertiary education, and 61% declare a predominantly scientific culture.

### 3.2. Purchasing Habits

Regarding shopping habits, the survey asked respondents whether they pay attention to the label, expiration date, and other indications on the packaging when making a purchase. The possible answers were always, often, sometimes, rarely, and never. The survey results show that the sample interviewed always pays attention to what is stated on the label (65%) or does it often (24%).

Furthermore, the questionnaire asked the consumer which information reported on the label was most important, such as expiry date, ingredients, geographical origin, brand, etc. The consumer could indicate a maximum of 3 preferences. Results show that respondents are more careful about the expiry date (87%), the ingredients (62%), the geographical origin (50%), and the presence or absence of some compounds in the ingredients (37%). Finally, a significant share of responses indicates attention to organic products (20%) and to the brand (20%).

Regarding factors influencing the food product choice, survey participants rated quality, price/quality ratio, ingredients, price, origin, promotions, and brand on a Likert scale ranging from 1 (not important at all) to 7 (very important). Table 1 shows the frequencies of the scores assigned for each factor, their means, standard deviations (SD), and medians.

Comparing the average scores obtained for each factor, quality, price/quality ratio, and ingredients obtain the highest average scores (6.4, 6.1, and 5.9, respectively). The brand obtains the lowest average value (4.1). However, the analysis of the frequency histograms of the scores assigned by the respondents shows the following (Figure 2):

For quality, quality/price ratio, and ingredients, there is an imbalance clearly in favor of the highest scores 6–7; i.e., most consumers find these factors important or very important.

The origin has a distribution like that of the three most favored factors, but with much lower frequencies of the highest score. The origin, however, is an overall rather important factor (average value 5.2).

The price (average score 4.8) is important and very important (score 5–7) for just over half of the sample surveyed (265 responses), while it is not at all or not very important (score 1–3) for 167 consumers.

Promotions and brand show mirrored distributions. Promotions have the most selected score, 5–7 (for a total of 282 responses), while the brand receives the most responses with a score below 4 (333 responses) and obtains the lowest average score (average value 4.1).

### 3.3. Propensity for Novelties in the Food Field

The propensity to try new foods receives an average score of 5.2 on a scale ranging from 1 (strongly disagree) to 7 (strongly agree). On the contrary, the participants disagree or partially disagree with the assertion of distrust of novelties (average score 2.6) (Table 2).

The frequency distribution of consumers’ scores on the statements related to the propensity to novelties (Figure 3) shows that 68% of the participants assign a score of 5–7 (partially totally agree) to the item “I like trying new foods”. On the other hand, only 15% of the respondents partially strongly agree with “I do not trust novelties”.

### 3.4. Degree of Knowledge of Biotechnologies and Their Perception

Given the preponderance of consumers with university and post-graduate education in the sample interviewed, as well as scientific culture, we believe it is interesting to investigate any differences in the consumer’s approach to (new) biotechnologies based on the level of education and type of culture. To this end, we divided the participants into three groups: A (middle school or high school diploma; 173 participants), B (bachelor’s or master’s degree, PhD—humanities culture; 105 participants), and C (bachelor’s degree, master’s degree, PhD—scientific culture; 233 participants). Thus, we re-analyzed the results of some more significant questions for the objectives of our research based on these three consumer populations.

The detailed analysis of the respondents of the three groups shows a perfect balance in terms of the size of the city of residence (about 50% of the participants in each group live in a large city). On the contrary, the age and gender distributions are different across groups. Chi-square test analysis with a 95% confidence interval demonstrates that Groups A, B, and C are related to the respondents’ age (*p*-value 0.001) and gender (*p*-value 0.005). Indeed, 67% of the women surveyed are in Group A, 75% are in Group B, and 58% are in Group C. Regarding age, Group C is younger than A and B (very similar), with 45% of respondents between 18 and 24 years old.

#### 3.4.1. Awareness of Biotechnology and More Reliable Sources of Information

Considering the entire sample interviewed, most consumers (73%) say they have heard of biotechnology in the food sector. Analyzing the answers by splitting the surveyed sample based on level and type of education, we obtain the results in Table 3.

Having heard of biotechnology in the food sector is independent of gender, but instead depends on the level and type of education, as it was legitimate to imagine (*p*-value 0.001). In fact, the percentages of “Yes” for each group are in the following order: C (tertiary education and scientific culture) > A (middle and high school diploma) > B (tertiary education and humanistic culture). Group A shows higher values than Group B despite its lower education level. A possible explanation is that in Group A, there may be a discrete fraction of high school graduates with a scientific focus.

Those who affirmatively answered that they had heard of biotechnology in the food sector were asked for their opinion on the use of biotechnology in this sector. A total of 64% of respondents are favorable toward it, provided that biotechnology can compensate for food shortages, intolerances, diseases, etc., or make production more sustainable. A total of 26% of respondents are unconditionally in favor of it. If we analyze this question considering the three groups, A, B, and C, we obtain the data reported in Figure 4. The scale used was as follows: 1 (contrary), 2 (neutral), 3 (favorable if biotechnologies compensate for food shortages, intolerances, diseases, etc., or make production more sustainable), and 4 (unconditionally favorable). The results show that the means of Groups A and B are significantly different (*p* < 0.001) from those of C for both questions. Group C, composed of consumers with a university level and scientific culture, is more aware of biotechnologies and is more in favor of them, although not scoring the maximum score for both questions (Figure 4).

The three groups of participants also differ in terms of the source from which they heard about biotechnologies (Figure 5). Group C clearly differs from A and B (which show very similar trends) in the percentage assigned to the item “In specialized journals” (64% against 20–25% of Groups A and B), “From friends and/or relatives” (13% against 31–32% of Groups A and B), and “TV/Radio” (16% versus 33–34% of Groups A and B).

Regarding the reliability of the information sources (Figure 6), the scientific community is the most reliable source on the subject for the three groups, followed by the public authorities. Differences between the three groups can be observed. Group C has more trust in public authorities than the other two groups (55% against 50–51% of Groups A and B) and believes more in the scientific community (88% against 69% of A and 79% of B).

Regarding the best description of food biotechnologies (Figure 7), the items “Application of scientific and engineering principles to treat biological material to supply goods and services” and “Human intervention to alter the final products of a natural production process” find the three groups in agreement and obtain 23–26% and 2% of preferences, respectively. This last item, which denotes a negative interpretation of biotechnology, is therefore considered valid only for a negligible minority of the groups. The distribution of responses among the other items follows a similar trend among the three groups, except for the phrase “Techniques which artificially induce changes in the structure and function of a living organism or biological process for a purpose of concrete utility”. For this item, Group B gives a much lower percentage of answers than A and C. Moreover, almost half of Group C prefer the item “They use living organisms to obtain products, improve plants and animals”.

#### 3.4.2. Consumer Opinions on Biotechnologies and Their Uses

The questionnaire follows with the question relating to the current uses of biotechnology. According to consumers, biotechnologies are currently used, in order of importance, to improve the resistance of plants to parasites (77%), to improve the characteristics of food products (59%), to introduce nutrients into widely consumed foods (45 %), to increase the shelf life of foods (36%), for the leavening of bread and the fermentation of beer (27%), and for the production of yogurt and cheese (23%). Some consumers have the wrong conception of biotechnologies, as they believe they can be used to obtain larger animals (23%) or for cryogenics (5%).

Regarding the usefulness, safety, and moral acceptability of some uses of biotechnology, the participants assigned scores from 1, corresponding to the negative judgments of useless, risky, and morally unacceptable, and 3, related to the positive opinions (helpful, safe, and ethically acceptable). The value 2 indicated the uncertainty. As shown in Figure 8, the following can be seen:

The three groups judge the 2–5 uses helpful, with average values close to the maximum score. Regarding Use 1, i.e., introducing human genes into animals, the mean values of A and B (about 2.2) significantly differ from those of Group C, which has values around 2.5.

Regarding the risky judgment, only Use 3 (genetic tests for diagnostics) shows values above 2.4 for Groups A and B, and close to 3 for Group C, indicating an opinion of moderate safety. Manipulating human genes (Use 1) and the transfer of genes into plants to make them resistant (Use 2) are judged to be much more risky than safe, showing low mean values. Relative to this use, Group C has significantly higher values than A and B. Use 5 obtains unanimous neutrality, while Use 4 shows A and C as neutral and B lower than C.

The “moral acceptability” judgment is almost unanimous for Use 3 (close to the maximum score) and is prevalent for Use 4, with a higher score for Group C. For Use 5, Group B has lower values than C, but however high. Conversely, the introduction of human genes into animals (Use 1) and the transfer of genes into plants to increase their resistance (Use 2) have low values for Groups A and B and values above neutral for Group C (significantly different from A and B).

In the following question, the survey asked participants how much they agreed with some opinions about biotechnology. Some of these items repeated concepts already expressed in the previous question, but the setting was different, as was the scale used, from 1 (completely disagree) to 7 (completely agree).

Statistical treatment conducted on the three groups’ responses led to the results shown in Figure 9. Groups A and B are almost indifferent to Opinion 1, “Modifying foods during production is harmful to health”, while Group C partially disagrees (significantly different from A and B). The three groups agree in expressing the need to deepen biotechnological knowledge in the food sector to understand its long-term effects (Opinion 2). Groups A and B agree on the need for scientists to clarify the risks/benefits of biotechnology in the food sector (Opinion 3). In this regard, Group C differs from the others by showing a lower score, towards only partial agreement with the statement. Groups partially agree that, if well used, biotechnology leads to high-value products (Opinion 4). Group C, however, differs from A, agreeing more with the statement. The three groups are equally neutral with respect to Opinion 5 “Biotechnologies are sometimes the only remedy for food problems”. The groups are neutral about “Biotechnologies increase productivity while respecting the environment” (Opinion 6). Here, C differs from B with a slightly higher score. Finally, they agree equally on the influence of scientific knowledge of new processes on the consumer (Opinion 7).

It is interesting to note that with respect to three out of seven topics proposed in this question, the three groups do not have significantly different opinions. In the remaining four questions, the differences are minimal in three cases and always concern Group C. It, in fact, differs as follows:

From Group A on Opinion 4 (if well used, biotechnology can add value to products).

From Group B on Opinion 6 (biotechnology increases productivity while respecting the environment).

From both Group A and Group B on Opinion 3 (Scientists need to be clearer about the risks/benefits of biotechnology in food).

This may perhaps be explained by a better average knowledge of the field that prompts Group C, on the one hand, to trust innovations more, and on the other hand, to clarify that scientists are not always able to provide certain clear and detailed information.

Finally, Group C differs greatly from both A and B in Opinion 1 “Modifying food during production is harmful to health”, showing that more awareness and more effective communication with people less knowledgeable about the topic is needed.

#### 3.4.3. Relevant Information for Choosing Biotech Foods

The survey asked how relevant some information was for consumers in choosing food obtained through biotechnology on a scale ranging from 1 (not at all) to 5 (essential). The results are shown in Figure 10.

Among the factors influencing the decision to purchase biotech products, the three groups attach great importance to the positive and negative effects on health and the environment (average value of 4.2). The opinions on scientists follow, with an average of 4.1, but with Group C, as compared to A, having more confidence in science.

Slightly less important is the technology used for production (average value 3.9). The opinions of friends/acquaintances and the internet/social networks are of little relevance, with Group C deeming them less relevant than Groups A and B.

### 3.5. Celiac Disease Awareness and Propensity to Purchase Products with “Detoxified” Gluten

The survey continues with a section relating to the knowledge of celiac disease, the acceptance of biotechnological methods to obtain products with gluten rendered harmless, and the propensity to purchase such products.

We asked all the participants questions on these issues, not just celiacs, to see if there is a difference in perception between those forced to buy gluten-free products and those who can choose to buy them. As already mentioned, nowadays, many people, despite not being celiac and knowing that gluten-free products are, on average, more expensive than those containing gluten, prefer the former as they consider them healthier. Having to evaluate the awareness of celiac disease, we also assessed it in non-celiac people.

#### 3.5.1. Celiac Disease Awareness

Survey results show that 19% of participants are directly affected by celiac disease or have celiac family members in their household.

A total of 79% have friends, acquaintances, or relatives who are affected by celiac disease and therefore are aware of the problems associated with this disease. This aspect is confirmed by the fact that 78% choose “It is a disease with a genetic predisposition, the causes of which are still debated, which can be kept under control by a gluten-free diet” as the definition that best describes celiac disease. However, there is a 20% share who confuse celiac disease with an intolerance or allergy.

Regarding its diffusion in Italy, consumers show a certain lack of information, independently from education: only 17% of the target choose the correct option (about 1%); 28% opt for “2–9%”, 27% arrive at “10–25%”, and 11% say “26–40%”.

As can be imagined, celiac participants are more aware than non-celiacs of what celiac disease is. However, they do not know how widespread celiac disease is in Italy.

Groups A, B, and C do not differ for all the questions in this section.

#### 3.5.2. Consumer Opinion on Celiac Products

Relating to the possibility of finding gluten-free products with the same qualities as those containing gluten, 53% think it is difficult, while 30% believe it is possible.

Regarding the degree of satisfaction with celiac products, 40% of consumers believe that gluten-free food has reached an acceptable quality level for consumption. However, 38% believe they cost too much. A total of 27% say that “something is missing, even if it does not look bad”. A total of 23% cut it short by saying they are glad they do not have this problem. Some consumers state it is the current trend for non-celiacs (18% of answers). Small shares of the sample interviewed assert they are more digestible (12%) or, on the contrary, contain unhealthy additives (13%). A total of 19% of consumers answered that they did not have a precise opinion.

No significant differences were found between Groups A, B, and C for all the questions in this section.

#### 3.5.3. Consumer Opinion on Using Biotechnology for Celiac Products

The last part of the section dedicated to celiac disease asks questions relating to the use of biotechnology in the specific case of gluten-free products.

Figure 11 illustrates the pie charts relating to the results of the questions submitted to the respondents.

For these questions, the chi-square test of independence (confidence interval = 95%) showed no correlation between the answers and the level and type of education, except for question e. In this case, (*p*-value of 0.001), the willingness to pay more for the product with detoxified gluten is related to belonging to the three Groups A, B, and C.

Furthermore, interesting results emerge from Figure 11.

Regarding the use of biotechnology to make gluten in products intended for celiacs harmless in some way, 81% of the sample is in favor. However, the percentage of consumers drops if you ask them if they are in favor of tasting it themselves (65%), since the number of uncertain respondents increases (33%).

Concerning the propensity to purchase such products with “detoxified” gluten, only 57% would buy it, 38% are doubtful, and 5% do not want to.

Regarding the possible price of products with detoxified gluten, 35% of consumers would pay 10–25% more, 27% would spend up to 10% more, and 7% would be willing to spend more than 25%. A total of 31% do not know.

If they were not celiac, 27% would still be willing to buy products with detoxified gluten, but only if they knew they were good for their health. A total of 24% would not buy them, but 20% would do so without condition. A total of 19% are uncertain, and 11% do not know.

## 4. Discussion

The online snowball sampling method made it possible to carry out our survey during a period of the pandemic crisis, which severely limited the development of research activities. This method is considered sufficiently reliable, produces moderate bias, and is especially helpful if it is difficult to reach the subjects of the survey [16,17,18].

An ad hoc panel of initial participants was selected. These initial participants were asked to share the link to the survey questionnaire with friends, relatives, colleagues, etc. Although many people still consider it inappropriate to use social networks to recruit participants for surveys and questionnaires, just as many publications attest to their ease of use and the possibility of reaching large numbers quickly. After all, on a social network, it is possible and relatively cheap and fast to publish and promote advertisements directed at a specific audience (characterized by region, age, or gender, for example) [19], whereas traditional methods, such as newspaper advertisements, flyers, letters, e-mail and word of mouth, are inadequate for recruiting hard-to-reach, homogeneous demographic groups for the chosen criterion, as well as often being slow and expensive [20]. For example, Facebook itself, being used little by the very young, conveys messages to a more adult and differently demographically characterized population than other social networks [21].

The results shown in Section 3.1, “Consumer Characteristics”, describe a sample of participants with a higher education level than the Italian average. In fact, according to the Italian National Institute of Statistics, in 2019, only 62.2% of the Italian population had at least one secondary education qualification between the ages of 25 and 64, and only 19.6% had a tertiary education qualification [22]. Furthermore, there is a predominance of respondents with a scientific culture (61%). Despite the use of private channels and social media, the level of education of the sample interviewed is in line with the profile of the user of the institutional channels we chose for the survey dissemination, i.e., ENEA (Italian National Agency for New Technologies, Energy and Sustainable Economic Development) social channels, FIDAF (Italian Federation of Doctors in Agriculture and Forestry), and Food Bank and Observatory on Dialogue in the agri-food sector.

Most participants (78%), regardless of their education level and type, show a good understanding of celiac disease and correctly correlate it with a gluten-free diet. However, a fair portion of the respondents (20%) still confuses it with an intolerance or allergy to gluten. Furthermore, the sample interviewed overestimates the percentage of celiacs in Italy. Only 17% of respondents gave the correct answer. Among those who answered correctly, there is a slight prevalence of respondents under 45 years of age.

This confusion mirrors the picture revealed by the Italian Celiac Association, a spokesperson association for patients and their requests, which has been battling misinformation about celiac disease since 1979. See, for example, the National Celiac Week, conducted to increase awareness and debunk the fake news that circulates too much in the Italian media [23].

The opinions of the participants on the ease of finding, the quality, and the cost of gluten-free products are also in line with national sentiment and, even more, with the data of Federconsumatori, an Italian non-profit association for the protection of consumer rights [24].

Regarding the knowledge and perception of biotechnology in the food sector, our surveyed sample differs by level and type of education. However, it does not vary depending on the size of the city of residence.

Although some studies have shown that socio-demographic characteristics are less and less relevant in the choice of food in developed countries [25], other research has shown the opposite. Sajdakowska et al. (2018) observed a significant influence of these characteristics on the acceptance of technologies used to nutritionally enhance grain products. In their study, less educated male participants and those from smaller cities and rural areas were more supportive of using technologies to improve food. Well-educated female participants and those living in larger cities were more reluctant [26]. Conversely, Azodi et al. (2019) found no association between education level and positive opinion about biotechnology [27]. Fernbach et al. (2019) argue that the more the extremism of the opposition to genetically modified foods increases, the more objective knowledge of the matter decreases. At the same time, self-perceived understanding increases [28]. On the other hand, many studies have shown that higher education levels lead to greater attention to health and better food choices [29]. This aspect can be decisive in the choice of gluten-free biotech products by female consumers, who, it should be underlined, are more affected by celiac disease than men. As proof of this, the annual report on celiac disease to Parliament estimates that in 2020, 70% of Italian celiacs were women [30].

Beyond the sometimes contradictory results of some studies, many authors find that high levels of education lead to more rational opinions, especially regarding the risks associated with new technologies, and show a positive attitude toward genetic technologies [31,32,33,34].

From the data in the literature, it seems that young people are more open to advanced technologies [35] and that men are more enthusiastic about new technologies [26]. Since Group C has a higher level of scientific education but is also younger and with a higher proportion of males than A and B, the results obtained could be the fruit of opposing or synergistic influences.

Overall, the literature shows that the acceptance of foods obtained with biotechnology is inextricably linked to the perception of risk and possible returns in terms of health and the environment. The perception of naturalness and disgust are also important factors [32,33,34,36,37,38]. We confirmed that biotechnologies’ acceptance increases if health and environmental benefits are recognized. Our results show that this phenomenon is independent of education level and gender.

It is interesting to note how our survey provides a very different picture from the situation revealed in Italy by the research by Bucchi and Neresini (2004) [12]. Our results show that consumer opinion has changed over 20 years on the usefulness of food biotechnology and moral acceptability. In 2003, the introduction of human genes into animals to produce organs or tissues for transplants and to transfer genes into plants to make them more resistant (Cases 1 and 2 in Figure 8) were considered, on average, not helpful; 52% of the 994 interviewed at the time declared these uses morally unacceptable. Public opinion has, therefore, changed in these aspects. However, the perception of the risk associated with biotechnology remains the same in 2020, the year of our study, as in 2003. In addition, the 2003 survey recognized the scientific community’s high reliability in genetic technologies but that it did not place much trust in public authorities. Our results testify that consumers in 2020 trusted scientists and authorities. The high trust in public authorities can be a phenomenon linked to the specific period of a health emergency, characterized by strong cooperation between Science and Politics.

## 5. Conclusions

This study aimed to evaluate consumer awareness and acceptance of the use of advanced biotechnologies, particularly to obtain gluten-free products.

It was directed at celiac and non-celiac consumers, as gluten-free diets have now gained increasing popularity among the general population, athletes, and patients with pathologies other than celiac disease.

The use of an online questionnaire, disseminated through telematic channels, allowed us to reach a fairly large sample of respondents, which we certainly would not have achieved with traditional administration methods, given the restrictions due to the containment of the COVID-19 pandemic.

Overall, 64% of our respondents favor food biotechnology, provided it has beneficial effects on health or the environment. Moreover, our results suggest that participants with an objective need for alternative solutions welcome biotechnologies and biotech products.

We observed that the perception of biotechnology differs according to the level and type of education. Participants with a scientific tertiary education recognize a higher helpfulness of genetic technologies for producing organs or tissues for transplantation, increasing plant resistance, or obtaining safe foods for people with food problems. They are also more confident in the safety of genetic technologies in diagnostics and in producing low-cost drugs from plants. On the other hand, higher education makes respondents more critical of the harmfulness to the health of modified foods.

Regarding gluten-free products, 65% of the survey participants would taste food made harmless through a biotechnological approach, and 57% would buy it, paying much more than usual gluten-free foods.

Our results show a change in consumer opinion about the usefulness of food biotechnology and its moral acceptability compared to 20 years ago. There is no doubt, however, that much still needs to be performed by the scientific community to disseminate information relating to genetic biotechnologies and the need for their use in many fields in an effective and usable way, thus raising the level of awareness of consumers and citizens in general.

A limitation of our study is the difficulty of interpreting the answers of the respondents, only considering a few variables such as the level and type of education, gender, and age. Individuals may respond differently depending on their background, religious beliefs, values, political orientations, social class, and surrounding situations. Several factors can therefore determine the acceptance of biotechnology.

The study of public opinion is, therefore, very complex and requires further investigation to effectively set up an awareness campaign on food biotechnology and its products deriving from it.

A strategic role is played by the media, especially those most easily accessible to people with lower or no educational qualifications who rely, much more than others, on the internet and social networks.

## Figures and Tables

**Figure 1 foods-12-01808-f001:**
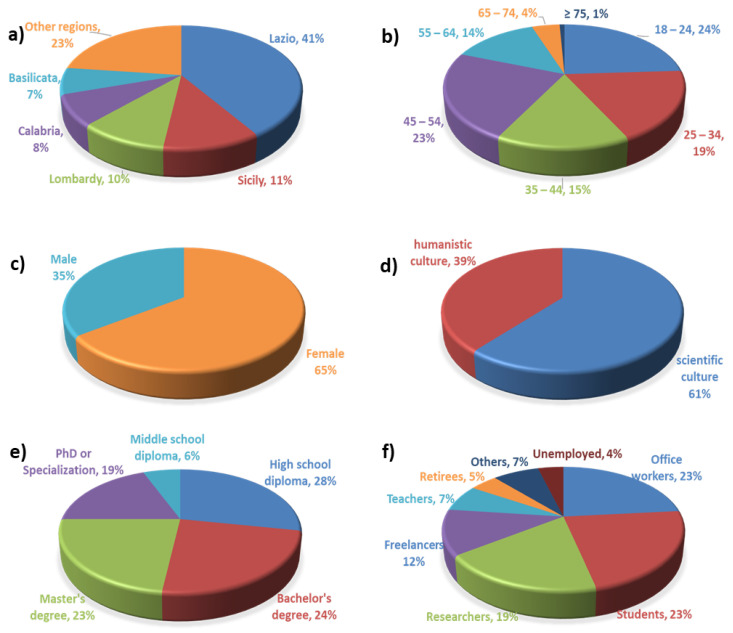
Main characteristics of survey participants: (**a**) origin, (**b**) age, (**c**) gender, (**d**) culture type, (**e**) education, (**f**) employment.

**Figure 2 foods-12-01808-f002:**
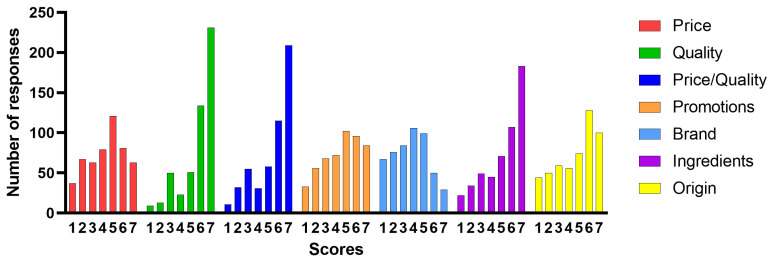
Frequencies of the scores for some factors influencing the food product choice. Scale used: 1 = not important at all; 2 = not important; 3 = partially not important; 4 = neutral; 5 = partially important; 6 = important; 7 = very important.

**Figure 3 foods-12-01808-f003:**
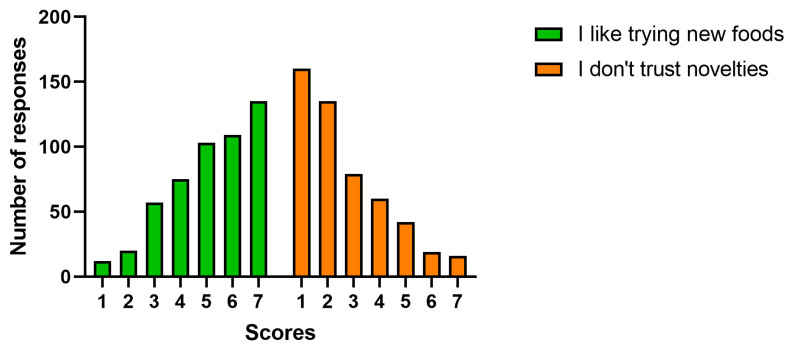
The propensity for novelties in the food field.

**Figure 4 foods-12-01808-f004:**
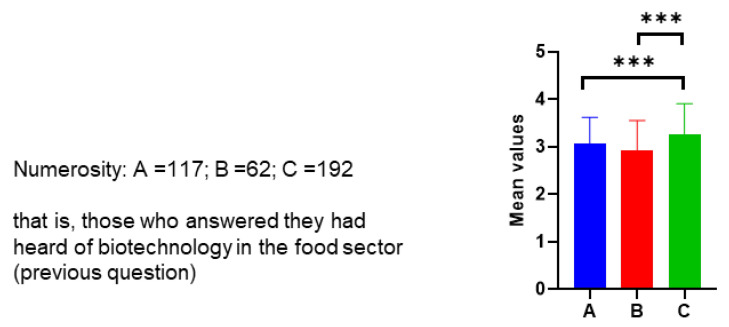
Responses relating the opinion on the use of biotechnology in the food sector for the three Groups A (middle school or high school diploma), B (bachelor’s or master’s degree, PhD—humanities culture), and C (bachelor’s degree, master’s degree, PhD—scientific culture). The scale used was as follows: 1 (contrary), 2 (neutral), 3 (favorable if), and 4 (unconditionally favorable). The post hoc Dunn test at the 5% level of significance was conducted. Significant differences were shown as asterisks on respective graphs, where *** *p*-value ≤ 0.001.

**Figure 5 foods-12-01808-f005:**
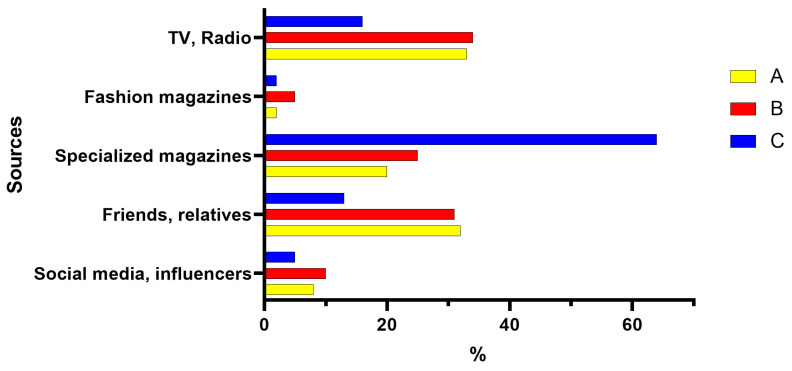
Sources from which Groups A, B, and C heard about biotechnologies. A (middle school or high school diploma), B (bachelor’s or master’s degree, PhD— humanities culture), and C (bachelor’s degree, master’s degree, PhD—scientific culture). Item percentages are calculated with respect to the total number of respondents in each group.

**Figure 6 foods-12-01808-f006:**
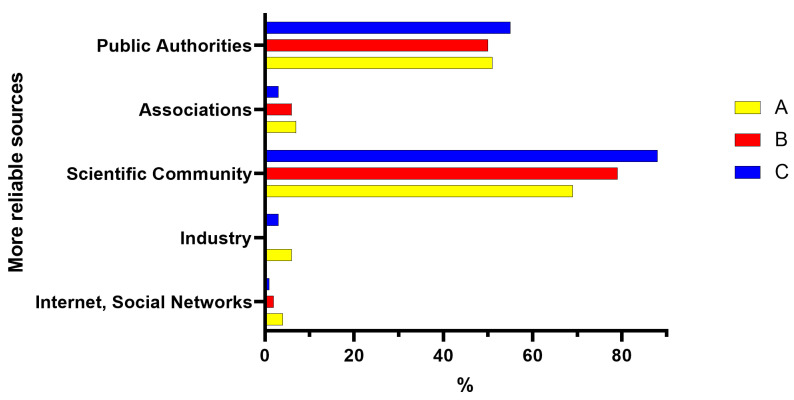
More reliable sources about biotechnologies for Groups A (middle school or high school diploma), B (bachelor’s or master’s degree, PhD—humanities culture), and C (bachelor’s or master’s degree, PhD—scientific culture).

**Figure 7 foods-12-01808-f007:**
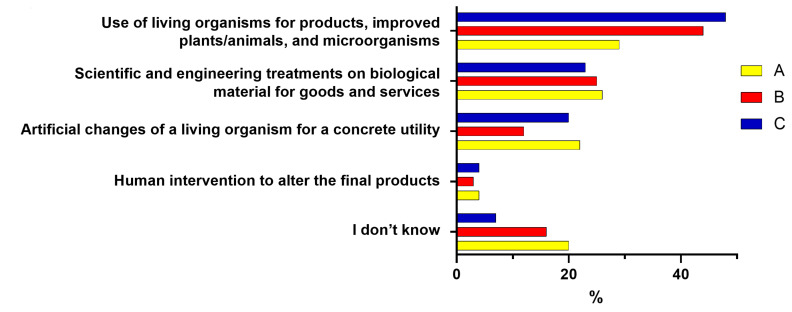
Responses (expressed in %) indicating the phrase that best describes food biotechnologies for the three Groups A, B, and C. A (middle school or high school diploma), B (bachelor’s or master’s degree, PhD—humanities culture), and C (bachelor’s degree, master’s degree, PhD—scientific culture).

**Figure 8 foods-12-01808-f008:**
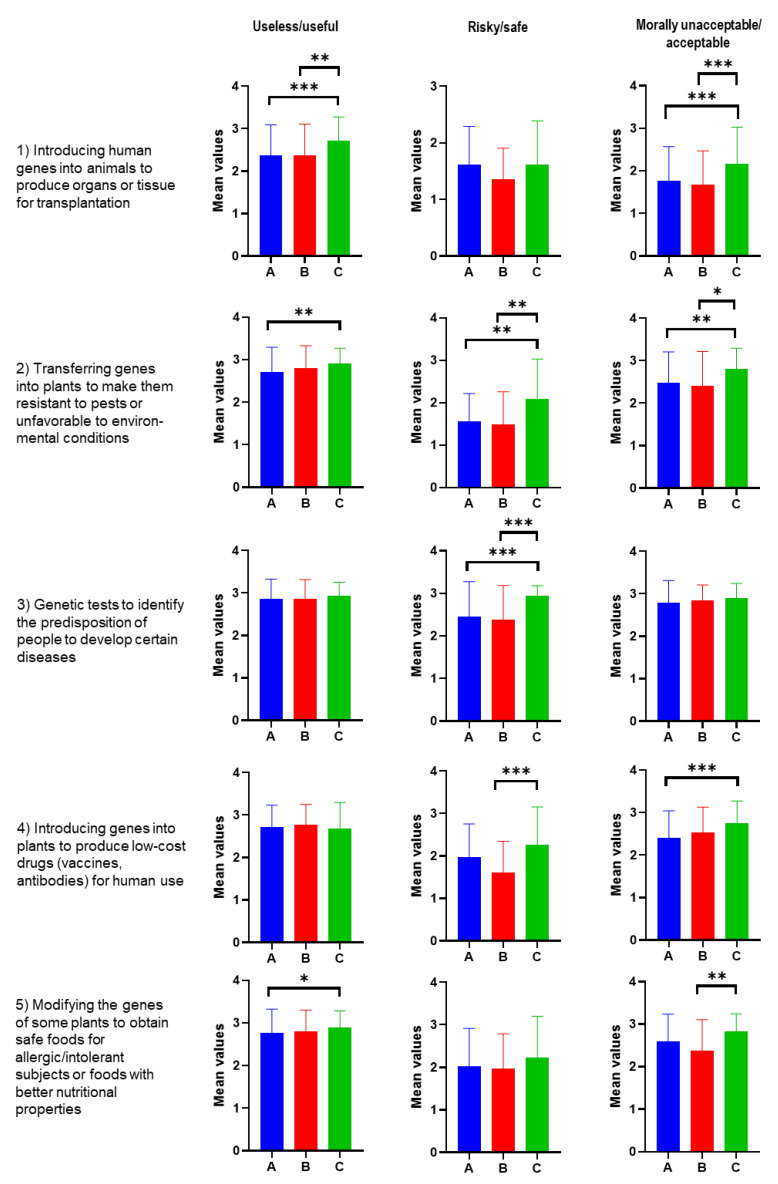
Responses about the usefulness, safety, and moral acceptability of some uses of biotechnology for the three Groups A, B, and C. Scale used: 1 (useless, risky, and morally unacceptable), 2 (neutral), and 3 (helpful, safe, and ethically acceptable). The post hoc Dunn test at the 5% level of significance was conducted. Significant differences were shown as asterisks on respective graphs, where * *p*-value ≤ 0.033, ** *p*-value ≤ 0.002, and *** *p*-value ≤ 0.001.

**Figure 9 foods-12-01808-f009:**
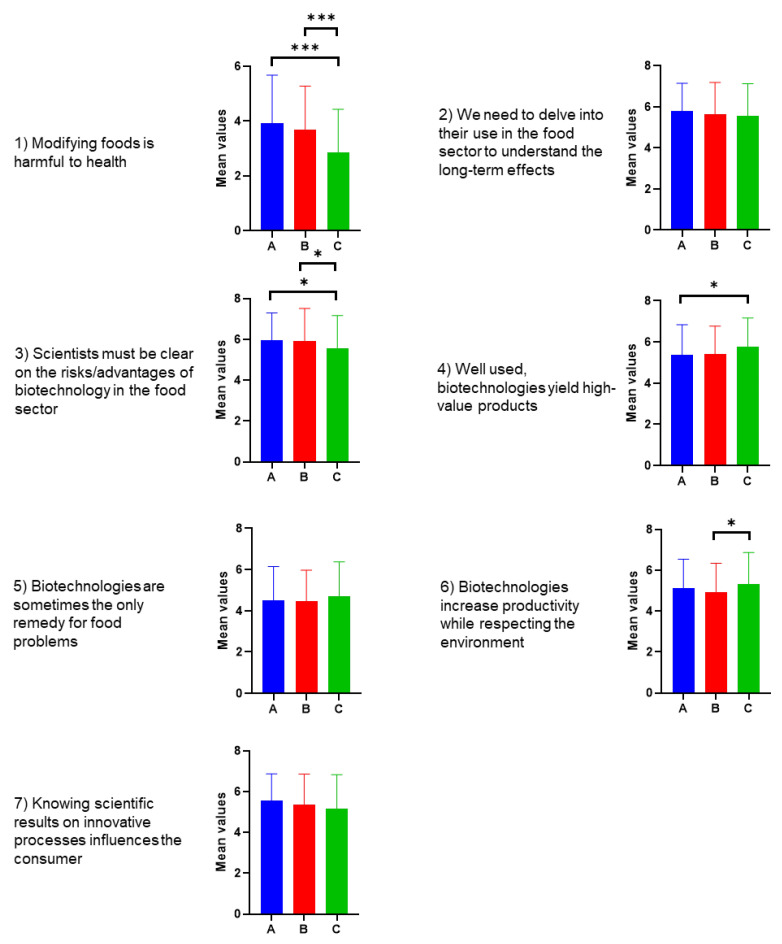
Responses relating the agreement with some statements regarding biotechnologies. The three Groups Are A (middle school or high school diploma), B (bachelor’s or master’s degree, PhD— humanities culture), and C (bachelor’s degree, master’s degree, PhD—scientific culture). Scale used: 1 = completely disagree; 2 = disagree; 3 = partially disagree; 4 = neutral; 5 = partially agree; 6 = agree; 7 = completely agree. Tukey’s test at the 5% level of significance was conducted. Significant differences were shown as asterisks on respective graphs, where * *p*-value ≤ 0.033 and *** *p*-value ≤ 0.001.

**Figure 10 foods-12-01808-f010:**
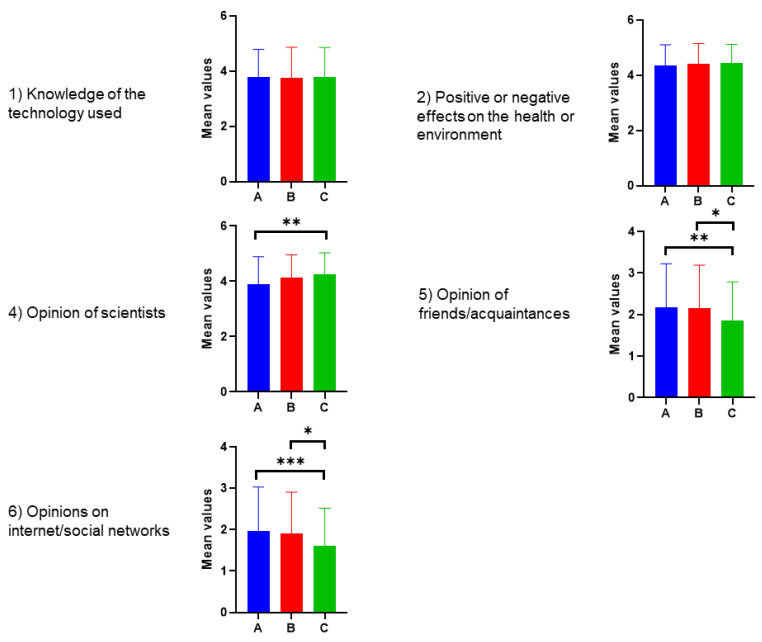
Responses of Groups A, B, and C to the question, “How important is the following information for you in choosing food obtained through biotechnology?”. Scale used: 1 = not important at all; 2 = of little importance; 3 = of average importance; 4 = very important; 5 = essential. Tukey’s test at the 5% level of significance was conducted. Significant differences were shown as asterisks on respective graphs, where * *p*-value ≤ 0.033, ** *p*-value ≤ 0.002, and *** *p*-value ≤ 0.001.

**Figure 11 foods-12-01808-f011:**
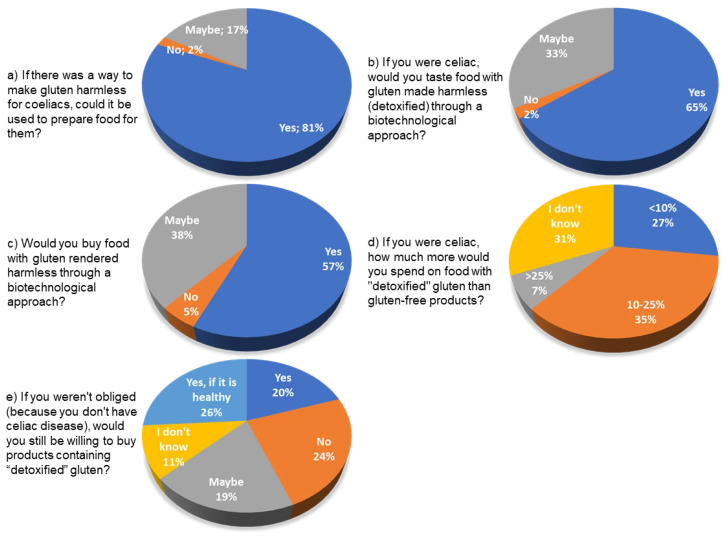
Pie charts relating the questions on the biotechnology use for gluten-free products.

**Table 1 foods-12-01808-t001:** Results relating to factors influencing the food product choice. Frequencies of the scores, mean score and standard deviation (SD), and the median. Scale used: 1 = not important at all; 2 = not important; 3 = partially not important; 4 = neutral; 5 = partially important; 6 = important; 7 = very important.

What Factors Influence the Choice of One Product Over Another Equivalent?	Scores							Mean (SD)	Median
	1	2	3	4	5	6	7		
Price	37	67	63	79	121	81	63	4.8 (1.8)	5
Quality	9	13	50	23	51	134	231	6.4 (1.5)	6
Price/Quality Ratio	11	32	55	31	58	115	209	6.1 (1.7)	6
Promotions	33	56	68	72	102	96	84	5.0 (1.8)	5
Brand	67	76	84	106	99	50	29	4.1 (1.7)	4
Ingredients	22	34	49	45	71	107	183	5.9 (1.8)	6
Origin	44	50	59	56	74	128	100	5.2 (1.9)	5

**Table 2 foods-12-01808-t002:** Results relating to the propensity to try food novelties. Frequencies of the scores, mean score and standard deviation (SD), and the median. Scale used: 1 = strongly disagree; 2 = disagree; 3 = partially disagree; 4 = neutral; 5 = partially agree; 6 = agree; 7 = strongly agree.

How Much Do You Agree with These Statements about the Propensity to Try Food Novelties?	Scores							Mean (SD)	Median
	1	2	3	4	5	6	7		
I like trying new foods	12	20	57	75	103	109	135	5.2 (1.6)	5
I do not trust novelties	160	135	79	60	42	19	16	2.6 (1.6)	2

**Table 3 foods-12-01808-t003:** Comparing knowledge of biotechnology based on level and type of education.

Have You Ever Heard of Biotechnology in the Food Sector?	Yes	No	Chi-Square Value	*p*-Value *
Group A	117	56	15.7	0.001
Group B	62	43		
Group C	196	37		

* *p*-value < 0.05 indicate that the considered variables are related.

## Data Availability

Data will be made available on request.

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
