# Peer review of "Consumer Awareness and Acceptance of Biotechnological Solutions for Gluten-Free Products"

_foods, 2023, doi:10.3390/foods12091808_

Round 1

Reviewer 1 Report

The manuscript describes original research aimed at identifying if there has been a positive trend in the perception of biotechnologies and, in particular, of foods produced with biotechnological approaches. The aim of the study does not collaborate clearly with the title of the study as stated. To generate reliable and trusted data to assess the awareness and acceptance of biotechnologically developed gluten-free diet for celiac disease patients, I would have expected mostly celiac patients to be involved in the study. This would have enhanced the impact of this manuscript.  As indicated by the authors, less than 19% of the 511 (i.e., less than 97 participants) as actual celiac patients. This is because any product that is developed will be primarily targeted to such patients. This is one of the main flaws of this manuscript. Altering the title to commensurate the content could make the data useful in supporting future studies.  However, authors will also need to improve on the readability of the manuscript before it is published.

The abstract will need reviewing to reflect actual results obtain in the study. It is not clear what authors mean by ‘The average consumer interviewed is female (65%)’. Another question is, is this the main answer to the aim of the study that should be presented in the abstract? Only this and, age and educational levels are the main results reported in the abstract. Authors also indicated in the abstract that the current studies evaluated consumer awareness of celiac disease impact; however, no data is presented to demonstrate consumer awareness  to that tune.

There are sections of the result presentation that will need re-writing for clarity. Most parts are unclear. E.g., Lines 166-184 – apart from education level that was used in the analysis, it is not clear how the other information provided helped in the interpretation of the results obtained. Similarly, lines 296-298, 299-303, 314-319,339-342, 348-351; just to mention a few will need re-writing for clarity. Throughout the manuscript, the word ‘sample’ has been used instead of ‘participants’. E.g., Lines 180, 183, 189, 217, 236 etc. Similarly, ‘interviewee’ and ‘interviewed’ are indicated when no interview was used in the study. E.g., Lines 183, 229, 243, 255. The word ‘regards’ as used in many cases, must be replaced with ‘regarding’: check lines 171, 178, 181, 186, etc.

In some cases, the results presented in the text does not collaborate with what is presented in the figures. E.g., line 301-303.

There is no clear distinction of what is presented in the discussion and the results. The discussion looks virtually like a summary of the results. I would suggest, authors combining the results and discussion.

I will suggest authors to include page ranges in all journals in the reference list, according to the journal requirements.

Finally, authors to ensure all acronyms or abbreviations are clearly defined when first used. E.g., ENEA

Reviewer 2 Report

Dissemination of the questionnaire via Facebook might cause a bias

Percentage and opinion of Celiac Patients should be highlighted

The Abstract and the Conclusions should be reconsidered by authors to include the outcomes of the different sections of the survey, which are overlooked

Screening outcomes should appear in the Abstract

Mention the need for awareness on biotechnology & GMOs 

Round 2

Reviewer 1 Report

Authors have made a good effort to revise the manuscript. Manuscript now has a clear theme in line with the title, aim and objectives, and improved readability. Just few suggestions below.

Line 21-22: ‘higher price than the market’. This is unclear: do authors want to mean higher price that the current market price of the product? Please, clarify or amend for clarity.

Line 146: ‘The patents mentioned above prove it.’ I am not sure if this is needed as providing reference citation supports the claim. Or just indicated ‘as demonstrated by the above studies’.

Line 442: Suggest deleting ‘full stop’ after (19%) and replace with ‘of which’.

Author Response

Thank you for your suggestions. We have modified the text according to your indications. 

Reviewer 2 Report

The quality of the manuscript has been improved. 

Many of the issues were addressed.

Still have to address the following (included in Report 1):

Unify the Figures format – Use the same template

Replace Questions Statement with explicit factors (Figures 7, 8, 9, 11)

Normalize the score axis scales

Add standard deviation bars where applicable 

Author Response

Unify the Figures format – Use the same template

We have uniformed the format of the figures, as you suggested.

Replace Questions Statement with explicit factors (Figures 7, 8, 9, 11)

Where possible, we have simplified some of the question statements, as you suggested. However, we have left figures 8, 9, and 11 unchanged because we believe reading the questions is more immediate, thus making it easier to understand the graphs.

Normalize the score axis scales

In each figure, the results are normalized and, therefore, comparable. The scale used is shown in the caption for ease of reading.
We believe there is no need to normalize the axis scales of all scores since we do not compare different figures' results.

Add standard deviation bars where applicable 

We have checked all the figures. For some figures, standard deviation bars are not applicable.